# Rhizobacteria Mitigate the Negative Effect of Aluminum on Pea Growth by Immobilizing the Toxicant and Modulating Root Exudation

**DOI:** 10.3390/plants11182416

**Published:** 2022-09-16

**Authors:** Andrey A. Belimov, Alexander I. Shaposhnikov, Tatiana S. Azarova, Darya S. Syrova, Anna B. Kitaeva, Pavel S. Ulyanich, Oleg S. Yuzikhin, Edgar A. Sekste, Vera I. Safronova, Margarita A. Vishnyakova, Viktor E. Tsyganov, Igor I. Tikhonovich

**Affiliations:** 1All-Russia Research Institute for Agricultural Microbiology, Podbelskogo sh. 3, Pushkin, 196608 Saint-Petersburg, Russia; 2Federal Research Center Vavilov All-Russia Institute of Plant Genetic Resources, 42–44, ul., Bol’shaya Morskaya, 190000 Saint-Petersburg, Russia; 3Department of Biology, Saint-Petersburg State University, University Embankment, 199034 Saint-Petersburg, Russia

**Keywords:** aluminum, immobilization, pea, PGPR, phosphorus, rhizosphere, root exudation, soil acidity, symbiosis

## Abstract

High soil acidity is one of the main unfavorable soil factors that inhibit the growth and mineral nutrition of plants. This is largely due to the toxicity of aluminum (Al), the mobility of which increases significantly in acidic soils. Symbiotic microorganisms have a wide range of beneficial properties for plants, protecting them against abiotic stress factors. This report describes the mechanisms of positive effects of plant growth-promoting rhizobacteria *Pseudomonas fluorescens* SPB2137 on four pea (*Pisum sativum* L.) genotypes grown in hydroponics and treated with 80 µM AlCl3. In batch culture, the bacteria produced auxins, possessed 1-aminocyclopropane-1-carboxylate (ACC) deaminase activity, alkalized the medium and immobilized Al, forming biofilm-like structures and insoluble phosphates. Inoculation with *Ps. fluorescens* SPB2137 increased root and/or shoot biomass of Al-treated plants. The bacteria alkalized the nutrient solution and transferred Al from the solution to the residue, which contained phosphorus that was exuded by roots. As a result, the Al concentration in roots decreased, while the amount of precipitated Al correlated negatively with its concentration in the solution, positively with the solution pH and negatively with Al concentration in roots and shoots. Treatment with Al induced root exudation of organic acids, amino acids and sugars. The bacteria modulated root exudation via utilization and/or stimulation processes. The effects of Al and bacteria on plants varied depending on pea genotype, but all the effects had a positive direction and the variability was mostly quantitative. Thus, *Ps. fluorescens* SPB2137 improved the Al tolerance of pea due to immobilization and exclusion of toxicants from the root zone.

## 1. Introduction

Aluminum (Al) is a common metal of all soils, but having high mobility in acid soils, it exerts phytotoxic effects on plants, resulting in the inhibition of growth and nutrient uptake [1,2,3]. Plants developed a number of tolerance mechanisms to toxic Al^3+^ ions, such as immobilization in the rhizosphere via increasing pH and production of mucilage and siderophores, root exudation of organic acids, sequestration and detoxification in plant tissues, efflux from roots and combating oxidative stress [4,5,6]. Pea (*Pisum sativum* L.) was considered as a relatively Al-sensitive plant species [7,8,9], but could counteract Al-induced oxidative stress [6,10,11] and immobilize Al in roots by pectin [12,13]. A comparative study of 106 pea genotypes revealed significant intraspecific variability in Al tolerance, and the proposed principal tolerance mechanisms in this species were the increase in the rhizosphere pH, Al precipitation in the root zone and maintenance of the plant nutrient homeostasis [14].

Legume plants form symbiosis with various types of microorganisms such as arbuscular mycorrhizal fungi (AMF), nitrogen-fixing nodule bacteria (rhizobia) and plant growth-promoting rhizobacteria (PGPR), which improve adaptation to abiotic stresses [15,16,17,18,19]. Positive effects of AMF on adaptation of various non-legume plant species to acid soils due to the formation of insoluble Al phosphates with mobilized P, improvement of nutrient uptake (Ca, Mg, K and Fe) and mitigating oxidative stress of plants were reported [20,21,22,23,24]. However, little is known about AMF effects on legumes grown in acid soils or treated with low pH and Al. Efficient nodule bacteria having high Al tolerance are present in acid soils [25,26,27]. Inoculation with such rhizobia increased nitrogen nutrition and adaptation of mung bean [28] and soybean [29] to Al toxicity. Increased root elongation and decreased Al content in roots of maize was observed after inoculation with P-solubilizing PGPR strain *Burkholderia* sp., most probably due to binding Al by released phosphates [30]. The Al-tolerant PGPR strain *Viridibacillus arenosi* IHBB7171 producing auxins and 1-aminocyclopropane-1-carboxylate (ACC) deaminase stimulated the growth of pea, but its effect on Al-treated plants or plants grown in acid soil was not studied [31]. Although information about the role of symbiotic microorganisms in adaptation of legume plants to acid soils and Al toxicity is limited, it might be assumed that symbiotic microorganisms may contribute to counteracting negative effects of these stress factors on plants.

Recently, we showed that pea genotypes VIR1903, VIR8473, VIR7307 and VIR8353 grown in acid Al-supplemented soil increased shoot and seed biomass after inoculation with consortium of AMF *Glomus* sp., nodule bacterium *Rhizobium leguminosarum* bv. *viciae* and PGPR *Pseudomonas fluorescens* strain SPB2137 [32]. Moreover, decreased concentrations of water-soluble Al in the rhizosphere and Al transport from root to shoot, as well as improvement of nitrogen fixation and uptake of nutrients by the inoculated pea plants were also observed. However, it was difficult to determine the role of each component of the consortium in the observed phenomena, since there were no options for inoculating plants with individual microorganisms. Therefore, the purpose of this study was to identify the contribution and mechanisms of the effect of *Ps. fluorescens* strain SPB2137 on the tolerance of these pea genotypes to Al^3+^ ions using a hydroponic system.

## 2. Results

### 2.1. Properties of Ps. fluorescens SPB2137

Strains *Ps. fluorescens* SPB2137 and SPB2137gfp produced similar amounts of phytohormones auxins (indole-3-acetic, indole-3-butyric and indole-3-lactic acids), but did not produce abscisic and salicylic acids (Table 1). Both strains were able to utilize ACC as a nitrogen source (Appendix A) and possessed similar ACC deaminase activity (Table 1). Both strains utilized all the tested organic acids, sugars (except melibiose) and amino acids (Appendix A) and no differences were found in the pattern of the utilized compounds between wild-type bacteria and SPB2137gfp (data not shown).

The Al supplemented to the un-inoculated tubes remained in solution, whereas in bacterial cultures, about 88% and 87% of Al was found in the residue at AlCl_3_ concentrations of 80 and 160 µM, respectively (Figure 1a). The presence of bacteria caused a significant increase in the pH of the solution up to 7.0 (Figure 1b). The residue of the inoculated tubes contained P and its amount increased with increasing concentration of supplemented Al (Figure 1c). The amount of other metals, capable of forming insoluble phosphates, in the residues of inoculated Al-untreated tubes was 4.1 ± 0.2 µg Ca and 0.8 ± 0.1 µg Mg per tube. The amount of Ca and Mg in the inoculated Al-treated tubes was similar to those mentioned above (data not shown). The number of bacteria in the inoculated tubes was similar at 0 and 80 µm Al, but decreased by about three times at 160 µm Al (Figure 1d).

In the presence of 80 µm Al, the bacteria (collected from the bottom of tubes) formed relatively large clumps (conglomerates), and single cells were also present (Figure 2c,d). At 160 µm Al, the clumps formed by the bacteria were significantly smaller (Figure 2c,d). In the absence of Al, clumps were not found, and mostly single cells were observed (Figure 2e,f). Quantification of the observed clumps in the microscope field of view (MFV) established significant differences in the number (Figure 3a) and size (Figure 3b,c) of clumps at different Al concentrations in the nutrient medium.

### 2.2. Growth of Plants and Bacteria in Hydroponics

Treatment with Al decreased root, shoot and total biomass of pea genotypes VIR1903 and VIR8473 (Figure 4). Shoot biomass of VIR7307 also decreased with Al treatment (Figure 4b), leading to a decrease in total plant biomass (Figure 4c). Growth of VIR8353 was not affected by Al (Figure 4). Inoculation with *Ps. fluorescens* SPB2137gfp eliminated the inhibiting effect of Al on root, shoot and total biomass of VIR1903 (Figure 4). The bacteria also stimulated shoot growth of Al-treated VIR7307, but had no effect on the biomass of VIR8353. The genotype VIR7307 had the highest biomass of control plants, whereas shoot and total biomass of other genotypes were rather similar (Figure 4b,c).

### 2.3. Development of Bacteria in Hydroponics

Strain *Ps. fluorescens* SPB2137gfp actively multiplied in the nutrient solution during plant cultivation, most probably using root exudates as nutrients, and increased the population from 10^5^ CFU mL^−1^ (initial bacteria concentration) to about 10^7^ CFU mL^−1^ with a maximum value in the presence of pea genotype VIR7307 (Figure 5a). The number of bacteria in the Al-supplemented solution was significantly less by about ten or one hundred times depending on pea genotype, and its number was bigger in the presence of VIR1903 as compared to VIR8473 and VIR7307. Roots of all pea genotypes were colonized by bacteria more actively in the absence of Al in the solution (Figure 5b). The highest colonization was found on roots of Al-untreated VIR1903 and VIR7307, whereas the lowest colonization was evident on Al-treated VIR8473 and VIR7307. No colonies of microorganisms were detected on BPF plates without antibiotics, suggesting the absence of contamination (data not shown). Characteristic patterns for colonization of root surface (Figure 6a) and root hairs (Figure 6b) by *Ps. fluorescens* SPB2137gfp are presented as an example using Al-treated pea genotype VIR1903. Bacteria are usually located and/or form micro-colonies along the junctions of epidermal plant cells, where leakage of root exudates probably occurred. No bacteria were found inside cells or tissues of the roots.

### 2.4. Distribution of Al in Hydroponics

Inoculation decreased the concentration of Al in the nutrient solution in pots where pea genotypes VIR8473 and VIR8353 were cultivated (Figure 7a) and increased the amount of Al in residues in growth media where any of the studied pea genotypes were cultivated (Figure 7b). Root Al concentration in VIR1903, VIR8473 and VIR7307 decreased in the presence of bacteria (Figure 7c); however, shoot Al concentration was not significantly affected (Figure 7d). Significant genotypic differences between peas in the ability to mediate the above-mentioned parameters were also observed (Figure 7).

Inoculation leaded to the increase in pH of nutrient solution where VIR1903 and VIR7307 were grown without the addition of Al, and the increased pH of Al-supplemented solutions was detected for any of pea genotypes (Figure 8a). Bacteria also increased the amount of P in the residue in growth media where Al-untreated VIR1903 and VIR8353 and all Al-treated pea genotypes were cultivated (Figure 8b). Genotypes VIR1903 and VIR8353 responded to Al treatment by the increase in the amount of P found in the residue.

Treatment with Al increased P concentration in roots of all inoculated and un-inoculated genotypes except VIR7307 (Figure 9a). Root P concentration of inoculated and un-inoculated plants was similar for any of pea genotypes; however, differences between control treatments were significant. Shoot P concentration was not affected either by Al treatment or bacteria (Figure 9b).

### 2.5. Root Exudation

Total root exudation of organic acids was significantly increased by Al treatment of all pea genotypes (Figure 10a). Inoculation had no effect on total organic acid concentration in the solution of Al-untreated plants, but decreased it by 2.4 times in Al-treated VIR8473 (Figure 10a). The major organic acid anions in root exudates of all pea genotypes were acetate, citrate, pyroglutamate and succinate (Appendix A). In the presence of bacteria, the solution concentration of acetate and succinate decreased in all pea genotypes, the concentration of citrate decreased in VIR7307, the concentration of lactate decreased in VIR7307 and VIR8353, and the concentration of pyroglutamate decreased in VIR8473, VIR7307 and VIR8353 (Appendix A). Trace concentrations of fumarate, malate and pyruvate were also detected, but the effects of Al treatment and inoculation were insignificant (data not shown).

Treatment with Al increased total exudation of amino acids by all pea genotypes (Figure 10b). Inoculation had no effect on total amino acid concentration in the nutrient solution in the absence of Al (Figure 10b). Opposite effects (the decrease or increase, respectively) of bacteria were observed on total amino acid concentration in growth media where Al-treated VIR1903 and VIR8473 were cultivated. Pea genotypes significantly differ quantitatively in the exudation of individual amino acids, and the observed differences varied depending on pea genotype and Al treatment (Appendix A). Qualitative genotypic differences were also observed. For example, VIR8353 did not exude Arg (Appendix A), but VIR8473 and VIR7307 did not exude Tyr (Appendix A). Characteristic features in the complex pattern of the observed effects were (i) an increase in the concentration of almost all determined amino acids due to the inoculation of Al-treated VIR8473; (ii) the most pronounced negative effect of bacteria on the concentration of amino acids in VIR1903; (iii) a significant increase in the concentration of many minor amino acids caused by bacteria in VIR7307, but no effect was evident on the total exudation by this genotype due to the 17-fold bacterial decrease in major component His; (iv) for the most part, Al treatment induced exudation of Asp, Glu, Gly, His, Pro and Thr (Appendix A).

Total exudation of sugars was significantly increased by Al treatment, particularly in genotypes VIR1903 and VIR7307 (Figure 10c). Bacteria decreased the concentration of sugars in the solution where these two genotypes were cultivated in the absence and presence of Al, and also in the case of Al-untreated VIR8353. Major sugar components in exudates of all pea genotypes were fructose and glucose, although relatively small amounts of ribose were exuded by VIR1903, VIR7307 and VIR8353 (Appendix A). Exudation all of these substances was positively affected by Al treatment, but the observed effects varied depending on pea genotype. In the presence of Al, bacteria significantly decreased the concentration of fructose and glucose (genotype VIR1903), glucose (genotype VIR7307) and ribose (genotype VIR8353) in the nutrient solution (Appendix A).

## 3. Discussion

### 3.1. Properties of Ps. fluorescens SPB2137

*Ps. fluorescens* strain SPB2137 was previously characterized as a PGPR protecting barley plants against infection by *Fusarium culmorum* [33]. Then, it was included into the microbial consortium containing AMF and rhizobia, and successfully used for the improvement of Al tolerance of the studied pea genotypes grown in acid soil [32]. To understand its plant growth promoting properties, we showed here that *Ps. fluorescens* strain SPB2137 produces phytohormones auxins and has ACC deaminase activity (Table 1). This strain utilized organic substances, many of which are components of the root exudates of peas [14,34,35,36,37] and other plants [38,39,40].

In batch culture, bacteria contributed to the transition of Al from the liquid phase of the nutrient medium to the residue (Figure 1a). This was accompanied by an increase in the pH of the medium (Figure 1b) and the accumulation of P in the residue (Figure 1c). Probably, part of Al precipitated in the form of phosphates, since the amount of P in the residue increased linearly with the increase in the amount of Al added to the medium. Assuming that Al_3_PO_4_ could be formed, this is supported by an approximate 3:1 molar ratio of Al to P found in the residue. For example, approximately 9 mg Al and 3.5 mg P were retained in the residue of the medium supplemented with 80 µm Al (Figure 1a,c). It is possible that a certain amount of Al was also adsorbed on or accumulated in bacterial cells. Removal of Al from the nutrient medium accompanied with the increase in P content in the residue was previously described for Al-tolerant PGPR *Curtobacterium herbarum* CAH5 [41]. The authors also observed morphological changes in microbial cells and suggested that Al was accumulated inside bacteria. PGPR *Bacillus megaterium* CAM12 and *Pantoea agglomerans* CAH6 [42] and yeast *Rhodotorula mucilaginosa* CAM4 [43] produced exopolysaccharides in the response to high Al concentrations in batch culture that could contribute to the formation of biofilm and protecting cells against Al toxicity [42]. Al-treated strain *Ps. fluorescens* ATCC 13,525 produced residue consisting of lipid moieties, Al and P [44], and P was involved in the accumulation of Al in these phospholipids [45]. In our experiments, *Ps. fluorescens* SPB2137gfp formed clumps in batch culture (Figure 2), which could be considered as a biofilm-like structures composed by cells, exopolysaccharides, phospholipids and Al phosphates. The formation of such structures occurred most actively in the presence of 80 µM Al (Figure 3). Significantly smaller clumps were observed at 160 µm Al and the number of bacteria in suspension also decreased, suggesting that clump formation was impaired and that this Al concentration was toxic.

### 3.2. Growth of Plants and Bacteria in Hydroponics

A previous report showed that pea genotypes VIR1903 and VIR8473 were less tolerant to Al toxicity as compared with VIR7307 and VIR8353 [14]. Here, this trait was preserved, since the growth of VIR1903 and VIR8473 was inhibited by Al to a greater extent than the other two (Figure 4). The positive effect of bacteria on plant growth was manifested to a greater extent in Al-sensitive genotypes. Al-sensitive plants were probably more dependent on interacting with bacteria to adapt to this stress. This speculation is related to a higher degree of symbiotrophy to AMF and rhizobia found in pea genotypes with reduced tolerance to Cd [46]. In that study, the conclusion was made that Cd-sensitive genotypes were more efficient in exploring the protective potential of symbiosis to compensate for their deficit in Cd tolerance.

The effect of *Ps. fluorescens* strain SPB2137gfp on pea biomass did not correlate with the abundance of bacteria in the solution or on roots. Therefore, the effect of bacteria was mainly due to their activity, which depended on the plant genotype properties, including those not associated with aluminum resistance. One of the common mechanisms of the positive bacterial effect was probably their ability to transfer Al from solution to residue, which manifested itself both in batch culture and in hydroponics. The amount of precipitated Al correlated negatively with its concentration in the solution (r = −0.43; *p* = 0.036; n = 24), positively with the solution pH (r = −0.43; *p* = 0.036; n = 24) and negatively with the Al concentration in roots (r = −0.43; *p* = 0.036; n = 24) and shoots (r = −0.43; *p* = 0.036; n = 24). Previously, we showed that the increase in rhizosphere pH and the exclusion of Al from the root zone are crucial mechanisms for the alleviation of Al toxicity in peas [14]. The involvement and important role of PGPR in these processes are shown here. As a result, inoculation decreased the Al concentration in pea roots, thus mitigating stress conditions. Our results are in agreement with the observed positive effect of Al-immobilizing PGPR *Burkholderia* sp. on maize [30], *C. herbarum* CAH5 on *Lactuca sativa* [41] and *B. megaterium* CAM12 or *P. agglomerans* CAH6 on *Vigna radiata* [42].

### 3.3. Distribution of Al in Hydroponics

The amount of Al precipitated in growth media with pea plants positively correlated with the amount of P in residue (r = +0.50; *p* = 0.013; n = 24) and P concentration in roots (r = −0.49; *p* = 0.18; n = 24), which also confirmed the formation of insoluble Al phosphates outside and inside the roots, respectively. It should be emphasized that P was not added to the nutrient solution (to prevent Al precipitation) and therefore, the plant root was the only source of this element in the solution. Al was likely precipitated as a result of the root exudation of substances containing P and originated from cotyledons. Indeed, treatment with Al increased the concentration of P in roots (Figure 9a). We hypothesize that root exudation of P is a protective response of plants to Al toxicity and can be considered as a mechanism aimed at immobilization and detoxification of this element. The presence of bacteria in the rhizosphere could activate the process of Al deposition by phosphates due to increasing pH of the medium. At the same time, shoot P concentration was not affected by Al treatment, suggesting no negative consequences for the growth of shoot, at least at this stage of plant development. The rest of the Al in residue could be associated with exfoliating root cells, as well as immobilized by bacterial cells or bacterial metabolites such as exopolysaccharides [42], phospholipids [44] and other compounds involved in interactions of microorganisms with Al [47]. The absence of the effect of bacteria on the shoot Al concentrations could be due to the relatively short period (10 days) of cultivation of the inoculated plants. However, a tendency to a decrease in Al concentration in shoots of VIR8473 genotype (*t* = 2.6; *p* = 0.12; n = 3) can be noted.

### 3.4. Root Exudation

Root exudation of organic acids is a well-known mechanism of Al tolerance in various plant species [48,49,50,51,52,53], including peas [14,54]. Consistently, aluminum treatment induced exudation of these substances in all pea genotypes (Figure 10a). The presence of *Ps. fluorescens* SPB2137gfp reduced the total concentration of organic acids only in VIR8473 (Figure 10a). The bacteria mainly utilized acetate and succinate exuded by all pea genotypes (Appendix A). The concentration of citrate, the dominant component and known as an active aluminum chelator, did not change in the presence of bacteria. This suggested that *Ps. fluorescens* SPB2137gfp utilized only specific organic acids in association with the plants that differ from its ability to use these compounds in batch culture (Appendix A). The obtained results also allow speculating that bacteria did not prevent the plants from chelating Al by organic acids.

Little is known about the effects of Al on root exudation of amino acids. Malate transporter TaALMT1, involved in Al tolerance of wheat, was also able to exude gamma-aminobutyric acid [55]. Roots of graminaceous plants respond to Fe deficiency by production of phytosiderophores related to non-protein amino acids belonging to the mugineic acid family [56]. Wang et al. [57] showed that wheat seedlings responded to Al toxicity by a significant increase in exudation of many proteinogenic amino acids, particularly Arg, Gly, Ile, Leu, Met, Phe, Pro and Val. However, their role in Al tolerance of plants was not investigated. In our study, three of four Al-treated pea genotypes exuded more amino acids estimated as the total amount of these compounds (Figure 10b). The pattern of major components in pea (Asp, Glu, Gly, His, Pro and Thr) shown in Appendix A partially differed from that described for wheat [57].

Significant genotypic variation in pea was found in exudation of the total amount (Figure 10b) and individual amino acids (Appendix A). The observed decrease in total amino acid concentration after inoculation of Al-treated VIR1903 could be explained by the utilization of these compounds by *Ps. fluorescens* SPB2137gfp. These results, confirmed by other reports, show active utilization of exuded amino acids by rhizosphere microorganisms [40,58,59,60,61]. It also was shown that bacterial metabolites inhibited amino acid uptake by roots of alfalfa, maize and wheat [61]. However, *Ps. fluorescens* SPB2137gfp increased total amino acid concentration of Al-treated VIR8473 by three times (Figure 10b) and also increased the concentration of several individual amino acids in growth media where any of the pea genotypes were cultivated with Al (Appendix A). This indicated that bacteria stimulated the exudation of amino acids by Al-treated plants. Stimulation of amino acid exudation by various plant species caused by root-associated bacteria was previously described [62,63,64]. However, the reason for such an effect was not investigated and the role of stress factors, including soil acidity and Al toxicity, was not studied. Here, we observed increased exudation of tryptophan by the inoculated plants treated with Al (Appendix A). This amino acid is a precursor for bacterial biosynthesis of auxins causing plant growth promotion by PGPR [65,66,67]. Auxins modulated signaling pathways involved in Al tolerance in plant roots [52], increased root exudation of organic acids [68] and alleviated Al-induced cell wall rigidity in the root apex [69]. Strain *Ps. fluorescens* SPB2137gfp produced auxins from tryptophan (Table 1) and probably could contribute to Al tolerance and promote pea growth using this compound exuded by roots. This assumption is confirmed by the fact that another auxin-producing strain, *Ps. fluorescens* 002, promoted the growth of maize seedlings germinated on filter paper and subjected to 90 μM of AlCl_3_ [70]. These results suggest that more attention should be given to studying the role of amino acids exuded by roots in plant responses to Al toxicity and acidic soil conditions. Modulation of amino acid exudation by microorganisms may not be limited to the use of these substances for nutrition, but participate in plant resistance to Al.

The increased root exudation of sugars in the presence of toxic Al was previously described in wheat seedlings grown in hydroponics, and the major component was glucose [57]. Here, three of four pea genotypes, particularly VIR1903 and VIR7307, responded to Al by the increased total amount of sugars (Figure 10c) consisting of fructose, glucose and ribose (Appendix A). The observed negative effect of *Ps. fluorescens* SPB2137gfp on the concentrations of sugars in the solution was probably due to the utilization of these substances as a carbon source (Appendix A). The results are in agreement with experiments with three soybean varieties also revealing genotype-dependent patterns for the utilization of exuded sugars (fructose, glucose and ribose) by PGPR *Ps. oryzihabitans* Ep4 [37]. However, significant genotypic differences modulated by Al suggest the complexity of nutritional interactions between plants and bacteria, when the plant should respond to both the micro-symbiont and the stress factor. The activity and functionality of bacteria can be largely determined by the availability of energy and carbon sources and the ability to use them efficiently. Therefore, interactions between plants and PGPR modulated by sugars in the presence of stress factors need more attention.

### 3.5. ACC Deaminase

Bacterial ACC deaminase plays an important role in the adaptation of plants to abiotic stresses, including toxicity of heavy metals [71,72,73,74]. However, information about the effects of ACC-utilizing PGPR on plants grown in the presence of toxic Al concentrations is limited. Thakur et al. [31] reported a positive effect of Al-tolerant and ACC-utilizing *V. arenosi* IHBB7171 on pea growth, but the plants were not treated with Al. Inoculation of *Trifolium subterraneum* and *Medicago polymorpha* with ACC-utilizing *Pseudomonas* sp. Q1 stimulated plant growth in the presence of high concentrations of Mn, which also is a toxic metal of acid soils [75]. However, the interaction between this strain and plants under high acidity or Al toxicity was not studied. In the present study, ACC was not detected in the root exudates of pea genotypes, regardless the presence of Al or bacteria. Therefore, either ACC deaminase was not involved in the effects of bacteria on plants due to the absence of this substance in the rhizosphere, or the bacteria utilized ACC directly from the roots. In our previous experiments with potato, ACC was detected in the rhizosphere, and inoculation with *Ps. oryzihabitans* Ep4, *Variovorax paradoxus* 5C-2 and *Achromobacter xylosoxidans* Cm4 decreased its concentration [46]. The modeling study of Glick et al. allowed proposing that bacterial ACC deaminase outcompetes plant ACC oxidase and actively supplies the bacteria with this substance without root exudation into the rhizosphere [76]. A more detailed study may provide an answer to the question about the involvement of ACC deaminase in the observed effects.

### 3.6. Significance of Factor’s Effects

Significance of the effects contributed by the factors (such as pea genotype, treatment with Al and inoculation with *Ps. fluorescens* SPB2137) on the studied parameters are briefly outlined in Appendix A. The contribution of the pea genotype was significant for all the parameters, suggesting an important role of genotypic features and properties in response to Al toxicity and interactions with bacteria. The treatment with Al also significantly contributed to all the parameters, except the content of P in shoots. This confirmed our observations that Al affected both plants and bacteria, but a relatively short exposition to Al probably did not allow influencing the transformation of P into shoot. The estimation of factor inoculation with *Ps. fluorescens* SPB2137 showed significant effects on plant growth, particularly on translocation of Al and P in the hydroponics (Appendix A). This highlights the significant role of bacteria in Al detoxification in the root zone. Root exudation of organic acids and sugars was also significantly affected by bacteria. The low value of criterion F in assessing the effect of bacteria on the exudation of amino acids was probably due to their opposite effect on 1903 and 8473 (Figure 10b). Interactions of these factors also were significant, but generally to a lesser degree as compared to the factors themselves (Appendix A).

## 4. Materials and Methods

### 4.1. Plants

Seed samples of four pea (*Pisum sativum* L.) genotypes, VIR1903, VIR7307, VIR8353 and VIR8473, were obtained from the N.I. Vavilov Institute of Plant Genetic Resources (Saint-Petersburg, Russian Federation) and multiplied for the experiments under similar soil and climate conditions.

### 4.2. Microorganism

PGPR strain *Pseudomonas fluorescens* SPB2137 was obtained from the Russian Collection of Agricultural Microorganisms (RCAM, St.-Petersburg, Russian Federation, http://www.arriam.ru/kollekciya-kul-tur1/; assessed on 3 March 2020). During the experiments, bacteria were maintained on agar Bacto-Pseudomonas F (BPF) medium [76]. Its derivative variant SPB2137gfp was marked with the gene encoding green fluorescent protein (GFP) [77], as described by Belimov et al. [40], and maintained on BPF medium supplemented with antibiotics rifampicin (20 mg L^−1^), gentamicin (15 mg L^−1^) and kanamycin (30 mg L^−1^). Fluorescent activity of SPB2137gfp was tested using a fluorescent microscope (Axio Imager A2, Carl Zeiss, Oberkochen, Germany). The properties of the wild-type *Ps. fluorescens* SPB2137 and SPB2137gfp were compared. Production of phytohormones (auxins, abscisic acid and salicylic acid) was determined using the UPLC system Waters ACQUITY H-Class (Waters Corporation, Milford, MA, USA), as described previously [40]. Activity of 1-amino-cyclopropane-1-carboxyllate (ACC) deaminase was determined by monitoring the amount of *α*-ketobutyrate (*α*KB) generated by enzymatic hydrolysis of ACC [78]. The protein concentration of cell suspensions was determined by the method of Bradford [79]. The ability of bacteria to utilize in vitro various substances (organic acids, sugars and amino acids) was determined by liquid batch culture supplemented with individual substances as a sole source of carbon or nitrogen and comparing turbidity with the inoculated medium containing no carbon source [37].

### 4.3. Bacterial Immobilization of Al in Batch Culture

Sterile test tubes were prepared (3 tubes for each treatment) containing 5 mL of nutrient medium (g L^−1^): sodium citrate, 5; KH_2_PO_4_, 1; NH_4_NO_3_, 0.25; MgSO_4_, 0.25; CaCl_2_, 0.1; pH = 5.6. The medium was supplemented with 0, 80 or 160 µM AlCl_3_ × 6H_2_O, and with *Ps. fluorescens* SPB2137 or SPB2137gfp in a final concentration 10^5^ cells mL^−1^. Tubes were incubated for 5 days at 25 °C in the dark without shaking. Aliquots of the sediment (residue) developed at the bottom of the tubes were gently taken by Pasteur pipette, avoiding agitation, and used for live microscopic observations. The presence of GFP-tagged bacteria in the residue was visualized using a laser scanning confocal microscope, LSM 510 META, and ZEN 2011 software (Zeiss, Oberkochen, Germany). GFP was excited at 488 nm, and fluorescence emitted between 501 to 554 nm was collected.

Then, the samples were vortexed and the number of bacteria was determined by the serial dilution method, as described previously [37]. The pH of the test tube contents was measured using the pH meter F20 (Mettler-Toledo, Schwerzenbach, Switzerland). The samples were centrifuged for 15 min at 9000× *g* and 4 °C. Residues were collected and digested in a mixture of concentrated HNO_3_ and 38% H_2_O_2_ at 70 °C using the digestion system DigiBlock (LabTech, Sorisole, Italy). Content of elements (Al, Ca, Mg and P) in supernatants (acidified with HNO_3_ up to a final concentration of 0.5% to prevent microbial activity) and in the digested residues was measured by an inductively coupled plasma emission spectrometer (ICPE-9000, Shimadzu, Tokyo, Japan), according to the manufacturer’s instructions.

### 4.4. Hydroponic Experiments

Pea seeds were surface-sterilized and scarified by treatment with 98% H_2_SO_4_ for 30 min, rinsed with tap water and germinated on filter paper in Petri dishes for three days at 25 °C in the dark. Seedlings were transferred to polypropylene pots (OS140BOX, Duchefa, The Netherlands) containing 250 mL of sterile nutrient solution (µM): KNO_3_, 1200; Ca(NO_3_)_2_, 60; MgSO_4_, 250; KCl, 250; CaCl_2_, 60; Fe-tartrate, 12; H_3_BO_3_, 2; MnSO_4_, 1; ZnSO_4_, 3; NaCl, 6; Na_2_MoO_4_, 0.06; CoCl_2_, 0,06; CuCl_2_, 0.06; NiCl_2_, 0,06. The nutrient solution was acidified up to pH = 4.7 via the addition of 1 M HCl and supplemented or not with 80 µM AlCl_3_ × 6H_2_O. The pH value and Al concentration were chosen based on previous experiments describing Al tolerance of the studied pea genotypes [14]. One pot with 10 uniform seedlings was prepared per genotype and treatment for each of the three independent experiments. Overnight culture of *Ps. fluorescens* SPB2137gfp was washed with sterile water by centrifugation at 9000× *g* for 5 min and the nutrient solution was supplemented with bacteria in a final concentration 10^5^ cells mL^−1^. Non-supplemented solution was used as a control treatment.

Plants were cultivated for 10 days in a growth chamber (ADAPTIS-A1000, Conviron, Isleham, UK) with 200 µmol of quanta m^−2^ s^−1^, a 12 h photoperiod and minima/maxima temperatures of 18 °C/23 °C. Then, root and shoot fresh weight (FW) of individual plants were determined and the pH of nutrient solution was measured as described above. The presence of GFP-tagged bacteria on roots was visualized using a confocal microscope as described above. The number of bacteria in the solution and on roots was determined by the serial dilution method using BPF medium supplemented with antibiotics [37]. The medium without antibiotics was used to check the presence of bacterial contamination.

The nutrient solution was centrifuged for 15 min at 9000× *g* and 4 °C, and 10 mL aliquots were taken for determination of Al concentration. The remaining supernatant was vacuum-filtered through 0.45 µm filters (Corning, Kaiserslautern, Germany), concentrated at 45 °C using a rotary vacuum evaporator (BUCHI R-200, BUCHI, Flawil, Switzerland) and used to determine root exudates (organic acids, sugars and amino acids) using the UPLC system Waters ACQUITY H-Class (Waters, Milford, MA, USA) as described previously [37]. Concentrations of Al in supernatants and element contents in the digested residues were determined as described above. Roots and shoots were dried at 50 °C, combined into one sample for each pot, ground into powder, digested and analyzed for element content, as described above.

### 4.5. Statistical Analysis

Statistical analysis of the data was performed using the software STATISTICA version 10 (TIBCO Software Inc., Palo Alto, CA, USA). MANOVA analysis with Fisher’s LSD test and Student’s *t* test were used to evaluate differences between means.

## 5. Conclusions

The studied rhizobacterial strain *Ps. fluorescens* SPB2137 produces auxins and possesses ACC deaminase activity, properties known as beneficial for host plants. These traits could be involved in adaptation of the inoculated pea genotypes to the presence of toxic Al concentrations, although more detailed study is needed to prove this hypothesis. One mechanism preventing the inoculated pea plants from Al toxicity was the exclusion of toxicants from the root zone due to the increase in the rhizosphere pH and immobilization of Al with phosphates. Bacterial cells and metabolites related to clump (probably biofilm) formation were likely also involved in the immobilization processes. Another mechanism could be related to the stimulation of Al-induced root exudation of P by bacteria, resulting in the formation of insoluble Al phosphates outside roots. This effect of PGPR on Al-treated plants is described here for the first time. The bacteria modulated root exudation of organic acids, amino acids and sugars via utilization as nutrients and/or stimulation of exudation processes. In this regard, it should be emphasized that inoculation of Al-treated plants maintained a relatively high concentration of organic acids to chelate the toxicant. For the first time, we showed the importance of amino acid and sugar exudation in interactions between PGPR and plants under stressful conditions caused by Al toxicity. Particularly, bacteria activated amino acid exudation and actively utilized sugars in the presence of Al. Whether these are positive effects of bacteria on plant adaptation to Al stress should be investigated in future studies. The observed interactions largely depend on the plant genotype, but no direct correlation between the growth response of plants to Al and the protective effect of inoculation was found. This indicates the complexity of the interaction between the components of the plant–microbial system and the environment.

## Figures and Tables

**Figure 1 plants-11-02416-f001:**
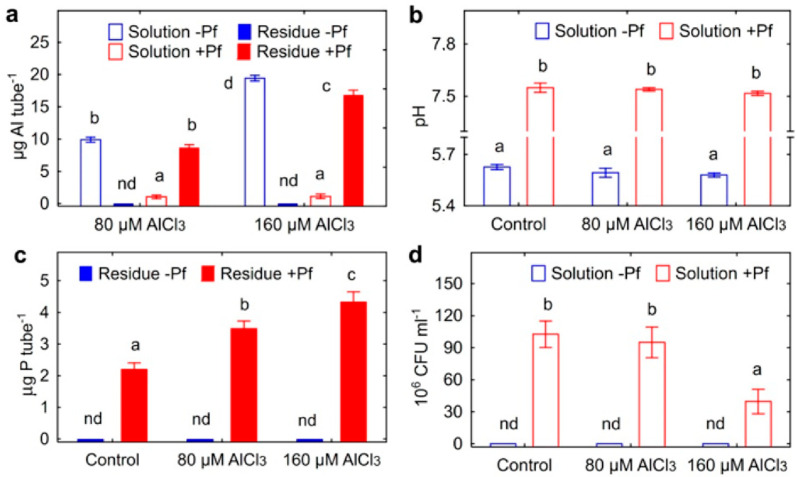
The amount of aluminum in solution and residue (**a**), pH of the solution (**b**), the amount of P (**c**) and the number of bacteria (**d**) in batch culture of *Ps. fluorescens* SPB2137 supplemented with 80 or 160 µM AlCl_3_. Treatments: −Pf—sterile control, +Pf—inoculated with *Pseudomonas fluorescens* SPB2137. Vertical bars show standard errors. Different letters show significant differences between treatments (least significant difference test, *p* < 0.05, n = 3). CFU stands for colony-forming units.

**Figure 2 plants-11-02416-f002:**
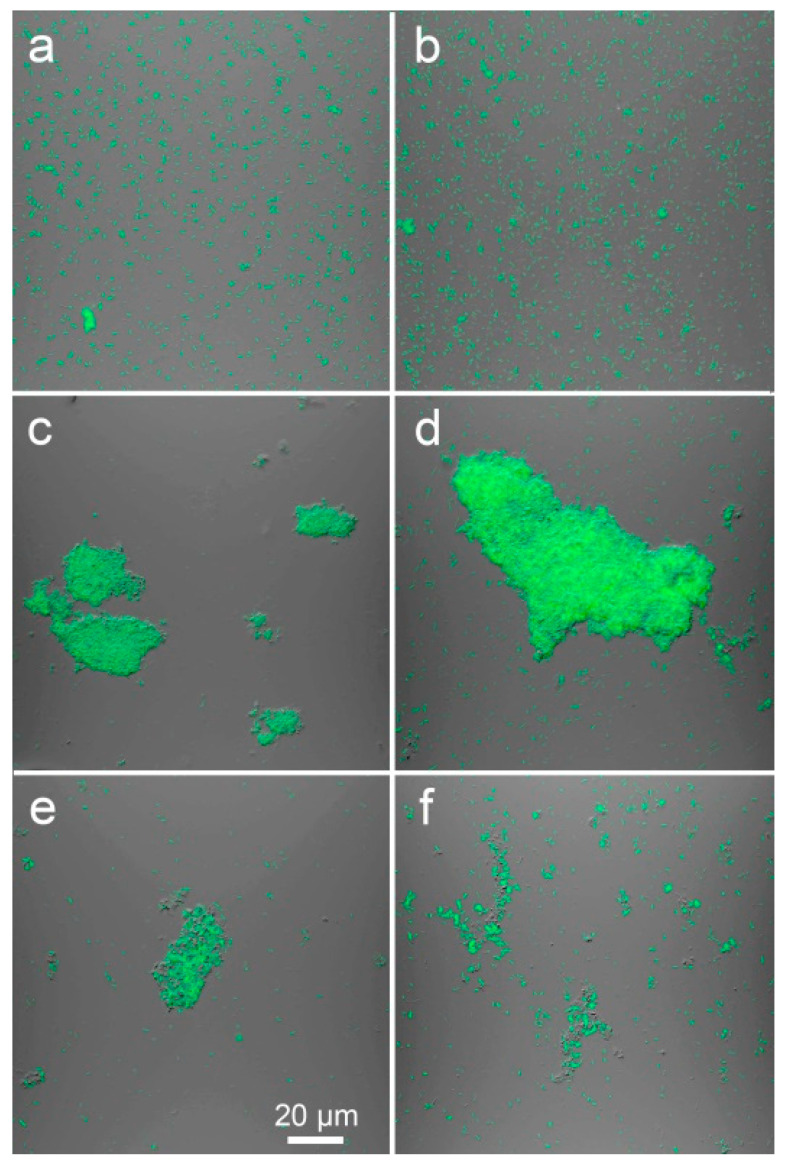
Confocal laser scanning images of *Pseudomonas fluorescens* SPB2137gfp batch culture in the absence of Al (**a**,**b**) and in the presence of 80 (**c**,**d**) or 160 (**e**,**f**) µM AlCl_3_. The bacteria are colored green. Images are single optical sections (**a**,**c**,**e**) and 10 merged optical sections (**b**,**d**,**f**). Scale bar (20 µm) shown in **e** is the same for all images.

**Figure 3 plants-11-02416-f003:**
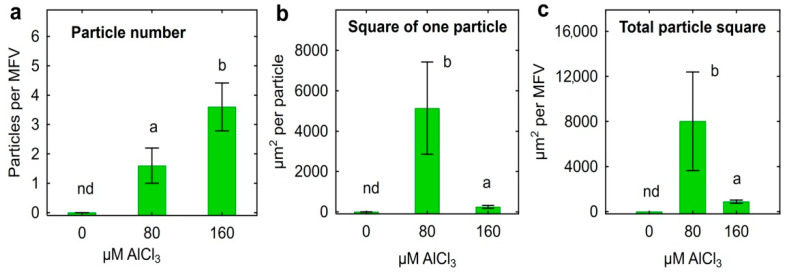
The number (**a**) and size (**b**,**c**) of clumps formed by *Pseudomonas fluorescens* SPB2137gfp in batch culture supplemented with 80 or 160 µm AlCl_3_. Vertical bars show standard errors. Different letters show significant differences between treatments (Student’s *t*-test, *p* < 0.05, n varied from 5 to 18 depending on the parameter and treatment). nd stands for not detected.

**Figure 4 plants-11-02416-f004:**
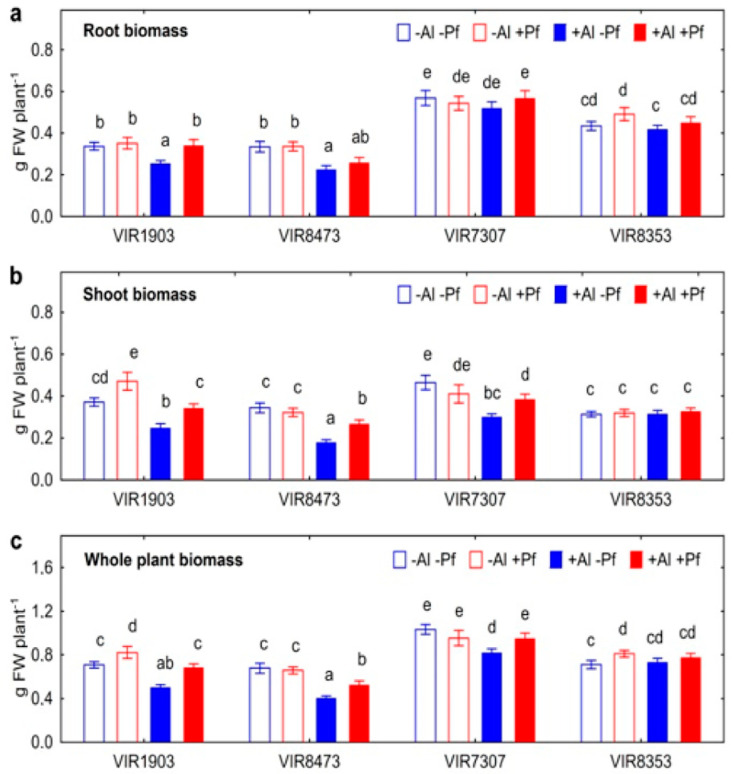
Root (**a**), shoot (**b**) and whole plant (**c**) biomass of pea genotypes VIR1903, VIR8473, VIR7307 and VIR8353 inoculated with *Pseudomonas fluorescens* SPB2137 and treated with 80 µm AlCl_3_. Treatments: −Al−Pf—Al-untreated and uninoculated control plants, −Al+Pf—Al-untreated and inoculated plants, +Al−Pf—Al-treated and uninoculated plants, +Al+Pf—Al-treated and inoculated plants. Vertical bars show standard errors. Different lowercase letters show significant differences between treatments (least significant difference test, *p* < 0.05, n varied from 16 to 24 depending on pea genotype and treatment). FW stands for fresh weight.

**Figure 5 plants-11-02416-f005:**
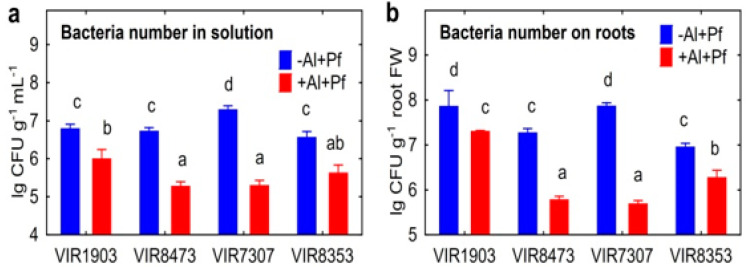
The number of *Pseudomonas fluorescens* SPB2137 in the nutrient solution (**a**) and on roots (**b**) of pea genotypes VIR1903, VIR8473, VIR7307 and VIR8353 untreated and treated with 80 µM AlCl_3_. Treatments: −Al+Pf—Al-untreated and inoculated plants, +Al+Pf—Al-treated and inoculated plants. Vertical bars show standard errors. Different lowercase letters show significant differences between treatments (least significant difference test, *p* < 0.05, n varied from 4 to 12 depending on pea genotype and treatment). FW stands for fresh weight. CFU stands for colony-forming units.

**Figure 6 plants-11-02416-f006:**
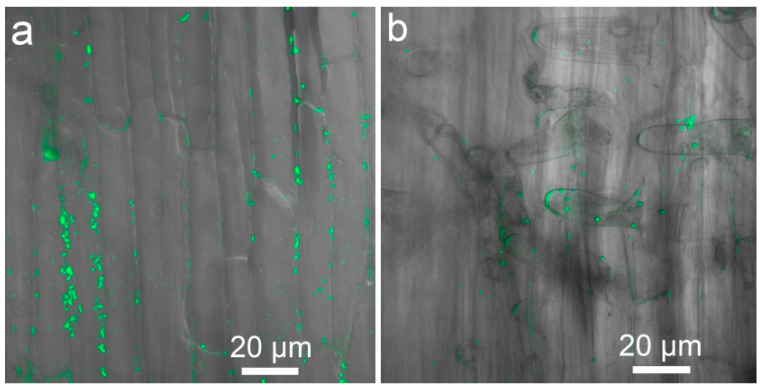
Confocal laser scanning images showing localization examples of *Pseudomonas fluorescens* SPB2137gfp attached to the root surface (**a**) and root hairs (**b**) of pea genotype VIR1903 treated with 80 µM AlCl_3_. The bacteria are colored in green. Images are 10 merged optical sections.

**Figure 7 plants-11-02416-f007:**
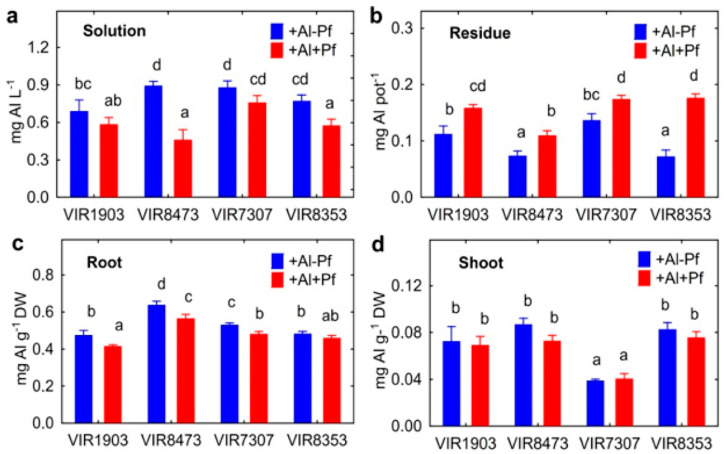
Effect of *Pseudomonas fluorescens* SPB2137 on Al concentration in the nutrient solution (**a**), aluminum amount in the residue in growth media (**b**) and aluminum concentration in roots (**c**) or shoots (**d**) of pea genotypes VIR1903, VIR8473, VIR7307 and VIR8353 untreated and treated with 80 µm AlCl_3_ in the end of experiments. Treatments: +Al−Pf—Al-treated and uninoculated plants, +Al+Pf—Al-treated and inoculated plants. Vertical bars show standard errors. Different lowercase letters show significant differences between treatments (least significant difference test, *p* < 0.05, n = 3). DW stands for dry weight.

**Figure 8 plants-11-02416-f008:**
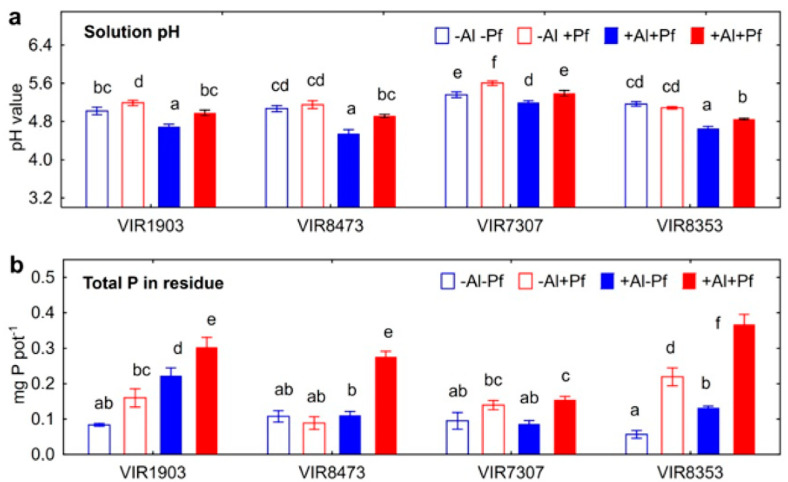
Effect of *Pseudomonas fluorescens* SPB2137 on solution pH (**a**) and amount of P in the residue in growth media (**b**) where pea genotypes VIR1903, VIR8473, VIR7307 and VIR8353 untreated and treated with 80 µM AlCl_3_ were cultivated. Treatments: +Al−Pf—Al-treated and uninoculated plants, +Al+Pf—Al-treated and inoculated plants. Vertical bars show standard errors. Different lowercase letters show significant differences between treatments (least significant difference test, *p* < 0.05, n = 3).

**Figure 9 plants-11-02416-f009:**
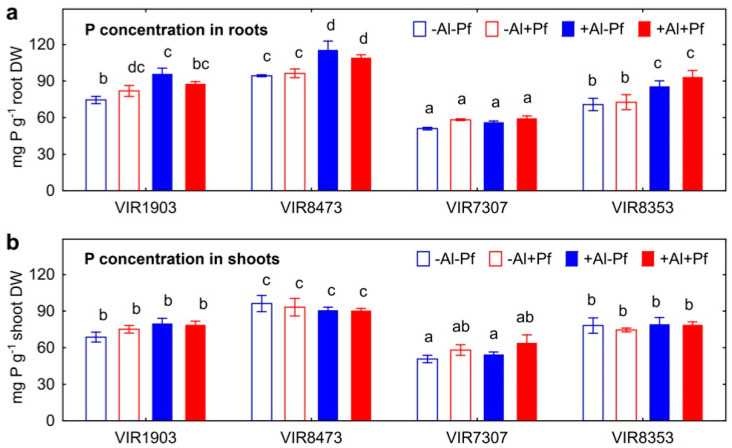
Effect of *Pseudomonas fluorescens* SPB2137 on concentration of P in roots (**a**) and shoots (**b**) where pea genotypes VIR1903, VIR8473, VIR7307 and VIR8353 untreated and treated with 80 µm AlCl_3_. Treatments: −Al−Pf—Al-untreated and uninoculated control plants, −Al+Pf—Al-untreated and inoculated plants, +Al−Pf—Al-treated and uninoculated plants, +Al+Pf—Al-treated and inoculated plants. Vertical bars show standard errors. Different lowercase letters show significant differences between treatments (least significant difference test, *p* < 0.05, n = 3). DW stands for dry weight.

**Figure 10 plants-11-02416-f010:**
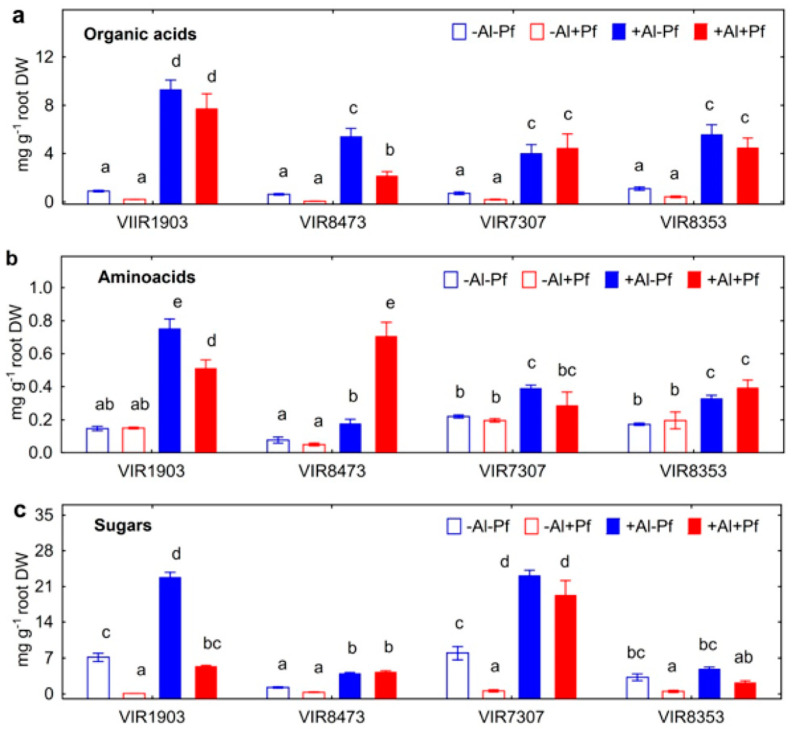
Effect of *Pseudomonas fluorescens* SPB2137 on total root exudation of organic acids (**a**), amino acids (**b**) and sugars (**c**) by pea genotypes VIR1903, VIR8473, VIR7307 and VIR8353 untreated and treated with 80 µM AlCl_3_. Treatments: −Al−Pf—Al-untreated and uninoculated control plants, −Al+Pf—Al-untreated and inoculated plants, +Al−Pf—Al-treated and uninoculated plants, +Al+Pf—Al-treated and inoculated plants. The data present the amount of exuded compounds of each category in sum for the period of plant cultivation divided by the root biomass. Vertical bars show standard errors. Different lowercase letters show significant differences between treatments (least significant difference test, *p* < 0.05, n = 3). DW stands for dry weight.

**Table 1 plants-11-02416-t001:** Properties of *Ps. fluorescens* SPB2137 and its GFP-tagged variant, SPB2137gfp.

Property	*Ps. fluorescens* SPB2137	*Ps. fluorescens* SPB2137gfp
Production of phytohormones (µ mL^−1^):		
Indole-3-acetic acid	572 ± 86 a	395 ± 48 a
Indole-3-butyric acid	1224 ± 183 a	1326 ± 48 a
Indole-3-lactic acid	41 ± 7 a	59 ± 6 a
Abscisic acid	nd	nd
Salicylic acid	nd	nd
ACC deaminase activity (µM αKB mg^−1^ protein h^−1^)	6.1 ± 0.4 a	5.8 ± 0.3 a

Data are means ± SE. Similar letter shows no significant difference between strains (Student’s *t*-test, *p* > 0.05, n = 2). nd stands for not detected.

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
