# Peer review of "Rhizobacteria Mitigate the Negative Effect of Aluminum on Pea Growth by Immobilizing the Toxicant and Modulating Root Exudation"

_plants, 2022, doi:10.3390/plants11182416_

Round 1

Reviewer 1 Report

Please review manuscript with attention to address small errors such as use of , instead of . , and misspelling such as antibiotivs instead of antibiotics. 

Can you please clarify units in figure 10? Units presented are mg/g root DW, but figure refers to exudation, which can bring some confusion. Did you calculated the concentration of organics (a, b, c), converted to total weight, sum all measured in each category (a, b, c) and after divided by the root DW? Maybe it worth mention in figure caption.

Would suggest the inclusion of a comment/hypothesis for the maintenance of concentration of Al in shoots under all the treatments studied and for all varieties (figure 7d), even if reduced nutrient solution and roots. What are the implications for human nutrition? 

Author Response

Dear Reviewer, the authors are very grateful to you for comments and suggestions. We tried to address all your comments. Our responses are in blue text below.

Reviewer 1

Comments and Suggestions for Authors

Please review manuscript with attention to address small errors such as use of , instead of . , and misspelling such as antibiotivs instead of antibiotics.

Response:

The manuscript has been checked for misspelling.

Can you please clarify units in figure 10? Units presented are mg/g root DW, but figure refers to exudation, which can bring some confusion. Did you calculated the concentration of organics (a, b, c), converted to total weight, sum all measured in each category (a, b, c) and after divided by the root DW? Maybe it worth mention in figure caption.

Response:

Yes, the data present the amount of exuded compounds of each category in sum for the period of plant cultivation divided by the root biomass. Correction has been inserted in figure caption (lines 260-261).

Would suggest the inclusion of a comment/hypothesis for the maintenance of concentration of Al in shoots under all the treatments studied and for all varieties (figure 7d), even if reduced nutrient solution and roots. What are the implications for human nutrition?

Response:

The following comment has been added on lines 378-381:

“The absence of the effect of bacteria on the shoot Al concentrations could be due to the relatively short period (10 days) of cultivation of the inoculated plants. However, a tendency to a decrease in Al concentration in shoots of VIR8473 genotype (t = 2.6; p =0.12; n = 3) can be noted”.

Al is not an essential trace element for humans, but it can be toxic to humans in high concentrations, and in our study the bacteria did not affect its entry into shoots. Therefore, we refrain from discussing human nutrition.

Reviewer 2 Report

The manuscript of Belimov et al. presents an interesting and actual topic; they studied the modulation of the impact of Al pollution on plants by beneficial bacteria. The aims and data are clearly presented, and the manuscript is described well. The authors should address, however several issues before the publication of their work:

Table 1 is of supportive only value, I suggest replacement into Supplementary material. In contrast, expressions like sugars, organic acids, and amino acids are very general; exact molecules should be mentioned. The data in S2 and S3 describe them in detail; still, they should be mentioned in the main text as well. The data are important and should also be mentioned in the main text.

The authors should edit statistical analyses to show the significance of the effect of bacteria, Al, or their interaction (genotype). The result and discussion part is nice since the authors avoided overwhelming the reader with descriptions of individual details. The general conclusions they made, however, need support on the effects of individual variables.

In Discussion the authors speculate bacterial biofilm is formed. I disagree considering bacterial clumps are formed. I can see the point of the authors, but they should use a more proper description of their suggestions.

In the Method part the authors claim that they checked for the presence of bacterial contamination in their bacterial sample. They either should explain how they did it (microscopically?), or the sentence should be skipped.

Minor issues:

The full name of Ps. fluorescens in the Abstract should be given as mentioned at the first time

Line 93: The authors claimed: that “in bacterial cultures about 90% of Al was found in the residue at both Al concentrations” – based on the figure this value is not valid/true for both concentration conditions; pls modify the value by min/at least etc.

Line 94: since not proved the cause of pH increase, the statement should be edited: “e.g., The presence of bacteria caused…”

Legend of Fig. 2: The bacteria colored… verb missing

The expression “(bacterial) particles” I consider misleading. The authors should use an alternative e.g., “clusters”, “clumps” etc.

Line 179: Al in residues in pots: pls consider instead: Al in residues in growth media

Unclear statements, which should be rephrased:

Line 92: The amount of Al supplemented to the un-inoculated tubes remained in solution (consider skipping “amount of”)

Line 254: Bacteria decreased concentration of 254 sugars in the solution of these two genotypes in the absence and presence of Al

Part between line 273-278 is misleading. Change in pH in accompanied/consequence of Al precipitation? How about the “timing” and effect of P exclusion? Please, carefully rephrase the part to avoid misleading or un-supported claiming!

Author Response

Dear Reviewer, the authors are very grateful to you for comments and suggestions. We tried to address all your comments. Our responses are in blue text below.

Comments and Suggestions for Authors

The manuscript of Belimov et al. presents an interesting and actual topic; they studied the modulation of the impact of Al pollution on plants by beneficial bacteria. The aims and data are clearly presented, and the manuscript is described well. The authors should address, however several issues before the publication of their work:

Table 1 is of supportive only value, I suggest replacement into Supplementary material. In contrast, expressions like sugars, organic acids, and amino acids are very general; exact molecules should be mentioned. The data in S2 and S3 describe them in detail; still, they should be mentioned in the main text as well. The data are important and should also be mentioned in the main text.

Response:

We believe that the characterization of bacteria presented in Table 1 is an important component of this paper that helps explain the effects produced by bacteria on plants. Previously, these properties in this strain have not been studied. In addition, these results justify the use of GFP tagged strain in the experiments. Therefore, we ask the reviewer to leave this table in the text of the article. These data may also be of particular interest to researchers studying interactions between PGPR and plants. The size of the article is not very large and it is quite possible to keep this table in the main text.

The data presented in Tables S2 and S3 are discussed in the text (see sub-section 3.4. on page 11). These detailed data are placed in the supplemental section so as not to overload the text of the article. So, we ask reviewer to leave these bulky tables as the supplemental material.

The authors should edit statistical analyses to show the significance of the effect of bacteria, Al, or their interaction (genotype). The result and discussion part is nice since the authors avoided overwhelming the reader with descriptions of individual details. The general conclusions they made, however, need support on the effects of individual variables.

Response:

To follow your comment a supplemental Table S6 and a subsection 3.6 have been added (lines 470-486). This material describes briefly the main effects of factors on the studied parameters related to plants and bacteria, as well as interactions between the factors.

In Discussion the authors speculate bacterial biofilm is formed. I disagree considering bacterial clumps are formed. I can see the point of the authors, but they should use a more proper description of their suggestions.

Response:

The term “biofilm” has been replaced by “biofilm-like structures” (lines 27, 322, 585)

In the Method part the authors claim that they checked for the presence of bacterial contamination in their bacterial sample. They either should explain how they did it (microscopically?), or the sentence should be skipped.

Response:

This sentence was located in a wrong place by accident. Now it has been transferred to the right place on lines 562-563.

Minor issues:

The full name of Ps. fluorescens in the Abstract should be given as mentioned at the first time.

Response:

Correction has been done (line 23).

Line 93: The authors claimed: that “in bacterial cultures about 90% of Al was found in the residue at both Al concentrations” – based on the figure this value is not valid/true for both concentration conditions; pls modify the value by min/at least etc.

Response:

This sentence has been corrected as you suggested (lines 97-98).

Line 94: since not proved the cause of pH increase, the statement should be edited: “e.g., The presence of bacteria caused…”

Response:

This sentence has been corrected as you suggested (line 98).

Legend of Fig. 2: The bacteria colored… verb missing

Response:

This sentence has been corrected as you suggested (lines 125-126).

The expression “(bacterial) particles” I consider misleading. The authors should use an alternative e.g., “clusters”, “clumps” etc.

Response:

The term “particle” has been replaced by “clump” throughout the manuscript (lines 129, 134-139, 321-326, 584-585).

Line 179: Al in residues in pots: pls consider instead: Al in residues in growth media.

Response:

In this sense, the word “pot” is everywhere replaced by “in growth media” (lines 196, 203, 214, 222, 268, 363, 415).

Unclear statements, which should be rephrased:

Line 92: The amount of Al supplemented to the un-inoculated tubes remained in solution (consider skipping “amount of”)

Response:

Correction has been done (line 96).

Line 254: Bacteria decreased concentration of sugars in the solution of these two genotypes in the absence and presence of Al

Response:

Correction has been done (lines 285-286).

Part between line 273-278 is misleading. Change in pH in accompanied/consequence of Al precipitation? How about the “timing” and effect of P exclusion? Please, carefully rephrase the part to avoid misleading or un-supported claiming!

Response:

The effect of bacteria on the pH of the solution as a result of the action of Al is unlikely, since the effect of bacteria was also in the absence of Al. However, the binding of Al by phosphates could affect pH. It is also likely that these processes are dynamic in time and their dynamics depends on the plant genotype. The sentence has been modified to avoid misleading (220-221). An explanatory sentence is also added on lines 379-382.
